# Klebsiella Lytic Phages Induce *Pseudomonas aeruginosa* PAO1 Biofilm Formation

**DOI:** 10.3390/v17050615

**Published:** 2025-04-25

**Authors:** Grzegorz Guła, Grazyna Majkowska-Skrobek, Anna Misterkiewicz, Weronika Salwińska, Tomasz Piasecki, Zuzanna Drulis-Kawa

**Affiliations:** 1Department of Pathogen Biology and Immunology, University of Wrocław, S. Przybyszewskiego 63/77, 50-148 Wrocław, Poland; grzegorz.gula@uwr.edu.pl (G.G.); grazyna.majkowska-skrobek@uwr.edu.pl (G.M.-S.); weronika.salwinska@uwr.edu.pl (W.S.); 2Department of Nanometrology, Wrocław University of Science and Technology, Z. Janiszewskiego 11/17, 50-372 Wrocław, Poland; tomasz.piasecki@pwr.edu.pl

**Keywords:** *Pseudomonas aeruginosa*, Klebsiella phages, non-specific viral–bacteria interactions, biofilm

## Abstract

Bacterial biofilms, characterized by complex structures, molecular communication, adaptability to environmental changes, insensitivity to chemicals, and immune response, pose a big problem both in clinics and in everyday life. The increasing bacterial resistance to antibiotics also led to the exploration of lytic bacteriophages as alternatives. Nevertheless, bacteria have co-evolved with phages, developing effective antiviral strategies, notably modification or masking phage receptors as the first line of defense mechanism. This study investigates viral–host interactions between non-host-specific phages and *Pseudomonas aeruginosa*, assessing whether bacteria can detect phage particles and initiate protective mechanisms. Using real-time biofilm monitoring via impedance and optical density techniques, we monitored the phage effects on biofilm and planktonic populations. Three Klebsiella phages, *Slopekvirus* KP15, *Drulisvirus* KP34, and *Webervirus* KP36, were tested against the *P. aeruginosa* PAO1 population, as well as Pseudomonas *Pbunavirus* KTN6. The results indicated that Klebsiella phages (non-specific to *P. aeruginosa*), particularly podovirus KP34, accelerated biofilm formation without affecting planktonic cultures. Our hypothesis suggests that bacteria sense phage virions, regardless of specificity, triggering biofilm matrix formation to block potential phage adsorption and infection. Nevertheless, further research is needed to understand the ecological and evolutionary dynamics between phages and bacteria, which is crucial for developing novel antibiofilm therapies.

## 1. Introduction

*Pseudomonas aeruginosa* is an environmental Gram-negative bacterium commonly found in soil and water, and it is also recognized as an opportunistic pathogen belonging to the ESKAPE multidrug-resistant bacteria. *P. aeruginosa* is associated with a wide range of severe infections, including wounds, burns, pneumonia, and sepsis [1]. One of the main virulence factors of *P. aeruginosa* is its capacity to develop intensive biofilms in response to harsh environmental conditions. These bacterial communities can be formed on a variety of surfaces, leading to various tissue and organ disorders [1,2]. A bacterial biofilm is a complex, organized multicellular structure of aggregated microorganisms adhering to a surface and surrounded by a matrix composed of extracellular polymeric substances (EPSs) [3]. The biofilm matrix is composed of polysaccharides, proteins, enzymes, lipids, signal molecules, and extracellular DNA (eDNA), which dynamically change in composition and amount to maintain a stable and integral structure, ensuring that the bacterial community adheres to particular surfaces [4,5]. The transition from planktonic to sessile form and vice versa is strictly controlled by quorum sensing (QS) and two-component (TCS) systems, enabling the fast modification of bacterial metabolism and production of fimbria, outer membrane proteins, secretion systems, and other virulence factors, in response to stressful conditions [6,7,8]. Biofilm milieu provides the opportunity for intense cell-to-cell interactions, facilitating horizontal gene transfer (HGT) and the accumulation of bacterial macromolecules [9,10,11]. In general, the biofilm protects embedded bacteria from unfavorable environmental factors, such as chemical compounds, dehydration, immune response, or widespread natural predators like bacterial viruses [2,12].

The insensitivity of sessile cells to antibiotic treatment forces us to look for alternative anti-biofilm strategies. One of the options is natural bacterial enemies, bacteriophages, which have high specificity to infect certain species or even strains of bacteria. Phages are currently intensively studied as potential antimicrobials to be applied in phage therapy, food production, agriculture, and biocontrol procedures [13,14]. Phages could also be considered antibiofilm agents since some of them can disrupt and degrade the biofilm matrix through the production of specific polysaccharide-degrading enzymes [15,16].

On the contrary, phage particles may also indirectly constitute a part of the bacterial community structure, leading to improvements in the spatial biofilm structure. Despite the common opinion that phages only destroy the biofilm structure, recent research shows that bacteria cells exposed to phages can significantly upregulate biofilm production [17]. A notable example is the temperate filamentous Pf4-like phage of *P. aeruginosa* (PAO1), which contributes to the proper spatial organization of the PAO1 biofilm. The Pf4-like phage propagates in sessile bacteria, resulting in cell lysis and the formation of channels within the biofilm structure, which is beneficial for surviving cells, even in multi-species biofilms [18,19]. Moreover, these filamentous phages form liquid crystalline complexes, encapsulating the bacterial host and non-specific bacterial cells, reducing their sensitivity to harsh conditions and stabilizing the biofilm structure [20,21]. It turns out that tailed phages may also constitute an important structural component of the multi-species biofilm composition in dental plaque [22].

Phage infection of bacterial cells leads to the production and release of various types of molecules. Due to the phage-derived cell lysis, the intracellular compounds can interact with the uninfected cells, significantly modifying their metabolism or even acting as signaling molecules for the bacterial community. Signaling agents that mediate specific and non-specific phage–bacteria interactions may include quorum sensing mediator particles, molecules encoded by the phage genome, membrane vesicles, or low-molecular-mass intracellular compounds from lysed cells [23]. These interactions may increase cell aggregation and biofilm formation [24].

The present study aimed to investigate the non-specific phage–bacteria interactions between the *P. aeruginosa* PAO1 population and Klebsiella phages. We monitored the PAO1 biofilm formation dynamics and planktonic culture density during co-incubation with phages by applying impedance and standard growth methods. Three Klebsiella phages characterized as myovirus (*Slopekvirus* KP15), podovirus (*Drulisvirus* KP34), and siphovirus (*Webervirus* KP36) were chosen, with the latter two equipped with virion-associated polysaccharide depolymerase. As the control, a lytic Pseudomonas phage (*Pbunavirus* KTN6) targeting LPS was used as the control of the specific phage impact on the biofilm formation. 

## 2. Materials and Methods

### 2.1. Bacterial Strains and Bacteriophages

The *P. aeruginosa* PAO1 (ATCC 15692) strain as the pseudomonas biofilm model was used in this study. Bacterial cells were maintained at −70 °C in a 20% glycerol-supplemented Trypticase Soy Broth (TSB, Becton Dickinson, and Company, Cockeysville, MD, USA). Before each experiment, the bacterial culture was rejuvenated on Trypticase Soy Agar (TSA, Becton Dickinson, and Company, Cockeysville, MD, USA) at 37 °C for 18–24 h. The refreshed bacterial culture was then diluted in physiological saline (PS, POCH SA, Gliwice, Poland) to an optical density (OD) of 1.0 (10^9^ CFU/mL) at λ = 600 nm and resuspended in 1.6 mL of TSB in a 24-well polystyrene culture plate (VWR International, LLC Radnor Corporate Centre, Radnor, PA, USA) to achieve a final density of 5 × 10^5^ cells per mL (CFU/mL) in each well. Four bacteriophages from the collection of the Department of Pathogen Biology and Immunology, University of Wrocław, Poland, were used as factors affecting the biofilm-forming activity of *P. aeruginosa* PAO1. We tested the impact of three Klebsiella phages, differing in taxonomy classification, virion morphology, genome size, and content, as well as recognizing different bacterial receptors: the myovirus *Slopekvirus* KP15 targeting outer membrane protein, podovirus (*Drulisvirus* KP34), and siphovirus (*Webervirus* KP36) both equipped with virion-associated K63 serotype capsule depolymerase. A lytic Pseudomonas phage (*Pbunavirus* KTN6) targeting LPS was used as the control (Table 1). Phages were propagated in TSB using their bacterial host strain. Briefly, the bacterial cultures in log-phase were inoculated phages at a virus-to-bacteria ratio (VBR) of 0.1–1.0 and growing at 37 °C with agitation (100 rpm). After the incubation, bacterial cultures containing phage particles were centrifuged (10,000× *g*, 20 min, 4 °C) and filtered (0.22 μm). Phage titer was assessed after serial dilutions in TSB followed by a double-agar overlay plaque assay [25].

### 2.2. Dedicated Setup for Bacterial Biofilm Measurement Using Impedance Technique

To examine the impedance spectra, the IMP-STM32 impedance analyzer was used and connected to a dedicated multiplexing measuring head with sockets for 24 QTF sensors (Figure 1). This setup allowed for the quasi-simultaneous measurement of impedance spectra measurements from 24 sequentially switched sensors in the frequency range of 100 MHz to 100 kHz in approximately 25 min intervals [30]. Then, 24 h before the experiments, the tuning fork heads were rinsed with isopropyl alcohol (Avantor Performance Materials, Gliwice, Poland) and placed in a titration plate with pure isopropanol for 15 min. Immediately before installing the sensors, the system was sterilized in a laminar flow chamber using ultraviolet for about 30 min.

Before the experiment, the enclosure was removed from the QTF sensors, which were sterilized with isopropanol. Alcohol was allowed to evaporate at room temperature. The system with QTF sensors was stored at 10 °C for an additional 24 h. During this period, the sensors were passivated (a process of thin-layer formation that stops the metal corrosion), which improved the measurement of impedance spectra of bacterial cultures. During passivation, the aluminum covers attached to the quartz sensor base become covered with oxides as a result of contact with the environment. This organized coating is resistant to further reactivity. This coating does not interfere with the adhesion of the tested bacteria to the sensor surface. To bring the sensing head and sensors to room temperature, the entire system within the airtight measuring cell was removed from the cold room 2 h before the bacterial measurement. Once the sensor temperature was equilibrated, a 24-well plate was placed and sealed to prevent contamination. For the duration of the experiments, the culture plate with the sensor head was kept in a measuring cell containing 200 mL of saturated potassium chloride solution (POCH SA, Gliwice, Poland).

The electrical Impedance Z = R + jX expressed In ohms (W) Is a complex measure of the object’s electrical response to the alternating voltage excitation, where R is resistance, X is reactance and j is the imaginary unit (a value expressed with a complex number). The technique used to measure impedance spectra, which represent these responses over a wide range of frequencies, is called impedance spectroscopy [31].

A typical impedance spectra analysis method involves fitting them with the impedance of the electric equivalent circuit (EEC) using dedicated software, for example, Scribner ZView 3.5i. In the case of the measurements of bacterial culture with QTFs as impedance sensors, it was noted that it was necessary to use two EECs (Figure 2a,b) as the electrical properties of the sensor change as the culture develops. Components of the EEC relate to the following: Rs, series resistance, which corresponds mainly to the growth medium resistance; CPEdl, electric double-layer capacitance between the growth medium and electrodes; Rct, charge transfer resistance; Rb and CPEb, resistance and capacitance of the objects adhered to the electrode surface (mostly cells and biofilm). The CPE is the element frequently used in EEC modeling as a generalized capacitor [31]. Its impedance is equal to Z_CPE_ = 1/Q(2πjf)^T^, where Q and T are CPE parameters and f is frequency. The EEC analysis of a sensor in which the EEC structure changes during the experiment is a complication, and the modeling itself requires specialized software. If the EECs are used and their correlation to the culture state is established, then it may be determined at which frequency in the spectrum a certain EEC parameter is the main factor determining the whole EEC impedance. Such essential EEC parameters were Rs and Qb (CPEb). The biological equivalent of changes for the Qb conductance is the adhesion of bacterial cells to the sensor surface as well as the formation and degradation of the biofilm EPS matrix. The analysis of the frequency ranges at which the most important EEC components influence the impedance spectrum shape allowed for simplification of the biofilm state assessment. Therefore, the Rs variations may be estimated by the real part of the impedance measured at 100 kHz. As the initial value of such a parameter varied between sensors, it was normalized by its value at the 4th hour of the experiment yielding R100k_norm. Similarly, the Qb variations were estimated by the variations in the conductance (that is, the real part of the inverse of the impedance) G100m measured at 100 mHz. The detailed spectrum measurement cascade, modeling using the RLC model, and simplified analysis at characteristic frequencies were presented by our team in our work from 2020 [32]. Measurement parameters, like high-frequency resistivity for (R100k) and low-frequency conductivity (G100m), were monitored. They were used to infer the behavior of planktonic forms (R100k) and the structure of the bacterial consortium (G100m), respectively [30]. All the details about the QTF impedance measurement compared to standard biofilm techniques can be found in the Appendix A.

### 2.3. Biofilm Dynamic Measurement Using Impedance Spectroscopy

A bacterial suspension with a density of OD_600_ = 1.0 was prepared from an overnight culture of the *P. aeruginosa* PAO1 strain in the TSA medium, which corresponded to 10^9^ CFU/mL. The starting suspension was then diluted to a CFU value of 10^6^ CFU/mL. The bacterial culture was combined with the phage lysate in VBR = 1, resulting in 1.6 mL of a bacterial culture at 5 × 10^5^ CFU/mL. A QTF sensor was immersed in the prepared culture to measure the impedance spectra (Figure 1). The experiment was carried out with *P. aeruginosa* PAO1 culture in the presence of Klebsiella phages KP15, KP34, and KP36. The PAO1-specific phage KTN6 was used as a positive control. During the impedance spectrum measurement, each data point was recorded at 25 min intervals and recalculated into resistance and conductance parameters, as previously described [30]. Analogous to the experiments conducted using bacteriophages, tests were carried out to determine the impact of phage lysate filtrate fractions below 30 kDa, 10 kDa, and 3 kDa respectively, on the dynamics of the biofilm. Fractions of phage filtrates were applied in a volume ratio of 1 to 1 (800 µL:800 µL) in the PAO1 strain system to the appropriate filtrate fraction. The resistance parameter R100k and conductance G100m corresponded, respectively, to the number of planktonic forms of bacteria and the growth dynamics of the biofilm on the surface of the QTF sensor. The spectra were recorded in 48 h experiments at 37 °C in TSB without replacement of the culture medium. Tests were performed at least in triplicate and 4 technical replicates. The sterility of QTF was monitored in the TSB medium.

### 2.4. Planktonic Culture Growth of P. aeruginosa PAO1 Measured by Optical Density

An 18–24 h rejuvenated culture of *P. aeruginosa* PAO1 was performed on the TSA medium. Then, a bacterial suspension was prepared with an optical density of 600 nm = 1 (10^9^ CFU/mL). The initial culture was then diluted in TSB to 10^6^ CFU/mL and mixed in the ratio 1 to 1 (100 µL:100 µL) with selected phage lysates (KP15, KP34, KP36, and KTN6). The OD_600_ measurements were made at time intervals every 30 min during a 24 h incubation at 37 °C in the 96-well flat-bottom microplate using Varioskan Lux multimode multi-plate reader (Thermo Fisher Scientific, Waltham, MA, USA). The experiments were carried out twice in technical triplicates.

### 2.5. Preparation of Bacterial Cell Lysates

Overnight *K. pneumoniae* cultures 77 (Kp77), 486 (Kp486), and ATCC 700601 (KpATCC) were diluted 1/100 in fresh TSB and cultured for 16 h at 37 °C, with shaking. After incubation, bacterial cells were kept at 4 °C for 3 h and subsequently disrupted using sonication (Sonopuls, Bandelin, Berlin, Germany). The whole-cell lysates were centrifuged (7500× *g*, 20 min, 4 °C). The resulting soluble fractions were filtered through a 0.22 µm filter (Millipore, Darmstadt, Germany) and stored at 4 °C for further analyses.

### 2.6. Preparation of Phage Lysate Fractions

Overnight cultures of *K. pneumoniae* strains were diluted 1:100 in fresh TSB and grown at 37 °C with agitation. After 2 h of incubation, bacterial cultures were infected with specific phages (KP15, KP34, KP36) at a VBR of 0.1 and further incubated at 37 °C for an additional 14 h. The cultures were then stored at 4 °C for 3 h. Following centrifugation (7500× *g*, 20 min, 4 °C) to remove bacterial cells, the supernatant containing phages was filtered through a 0.22 μm filter (Millipore, Darmstadt, Germany). The resulting phage lysates were processed using an Amicon Ultra-15 Centrifugal Filter Device with 30 kDa and 10 kDa molecular weight cut-off membranes (Amicon, Millipore, Darmstadt, Germany). The presence of phages in the fractions was verified using the spot test.

### 2.7. Adsorption Assay

Phage adsorption assays in homologous and heterologous systems were performed based on the protocol described by Bleackley et al., with some modifications [33]. Briefly, 3 mL of TSB containing approximately 10^8^ CFU/mL of log-phase bacteria and ~10^6^ PFU/mL of phage was incubated at 37 °C for 12 min without shaking. Samples were withdrawn at 4 min intervals (T = 4, 8, and 12 min) and centrifuged (1 min, 8000× *g*) to pellet bacteria with adsorbed phages. The supernatant was transferred to a new Eppendorf tube, serially diluted, and plated to determine phage titers. Phage adsorption was calculated as the percentage of unadsorbed phages, using the formula N/N_0_ × 100, where N_0_ represents the phage titer (PFU/mL) at T = 0, and N corresponds to the titer at each subsequent time point. Each assay was performed in duplicate.

### 2.8. Statistical Analyses

Statistical analysis was performed for each independent experiment in triplicate and four technical replicates for one measured test variant (N = 12). One-way ANOVA was used to compare differences between variances. Statistically significant differences in measurement variances were tested using Fisher’s post hoc tests (LSD, least significant difference, and Tukey’s HSD, significant difference test). Results were analyzed at the *p* = 0.05 level of significance.

## 3. Results

### 3.1. Dynamics of P. aeruginosa PAO1 Biofilm Formation Was Triggered by Non-Specific Klebsiella Phages

The dynamics of the *P. aeruginosa* PAO1 biofilm in the presence of non-specific Klebsiella phages (KP15, KP34, and KP36) and Pseudomonas lytic phage KTN6 were measured using impedance spectra. The results were analyzed based on a simplified analysis correlated with the detection of two electrical parameters. One, referred to as conductance in G100m, is equivalent to the biofilm growth on the surface of the sensor. Measurements of impedance parameters correspond well and are a reliable factor of the condition and dynamics of the biofilm, even during the phage treatment. The second parameter is the high-frequency resistance R100k corresponding to the propagation of planktonic bacteria. The conductivity parameter in G100m changed characteristically for the control experiment, with the PAO1 biofilm identifying the adhesion of cells to the sensor surface and the formation of a biofilm matrix (Figure 3), as demonstrated previously in our studies [30,31]. Changes in the G100m conductance dynamics were measured at 25 min intervals in a real-time automated manner without sample manipulation over 48 h of incubation.

By analyzing the curves and comparing the dynamics with the PAO1 control (green curve), an acceleration in biofilm formation after the application of non-specific phages was observed for 48 h. In the case of PAO1 growth control, an increase in conductance (G100m) was detected around the 10th hour of the experiment, corresponding to the adhesion of cells to the sensor surfaces and the formation of electrically conductive biofilm matrices. The correlation between biofilm initiation, mass formation, and conductance changes in G100m was described in our previous study [32]. The maximum conductivity of 700 µS for the control was observed around the 12th hour of incubation. Then, this parameter decreased about 6 h and increased again, suggesting that the fluctuations could be related to extracellular polymeric substance (EPS) component changes or partial biofilm degradation. The conductance dynamics for TSB (negative control) remained constant, maintained at a value of approximately 200 nS within 48 h. The application of non-specific phages resulted in a faster increase in G100m conductance for *P. aeruginosa* PAO1 biofilm, with changes between the 8th and 9th h of the experiment. Maximum conductivity was reached around the 10th hour for cultures with non-specific bacteriophages. Then, over the next 6 h, the G100m conductance decreased similarly to the course for the PAO1 control. The G100m conductance increased again at up to 1 µS and 600 nS for KP34 (16th–18th h), KP15, and KP36 phages (20th h). In the subsequent timing, this parameter declined until around the 36th hour of measurements, after which it slowly increased and stabilized. The electrical parameters of the biofilm did not differ significantly from the control, suggesting the fluctuations were probably related to nutrient depletion in the medium. The specific phage KTN6 used did not change the overall electrical character of the biofilm and the accumulated G100m parameter for the area under the curves between 4 and 24 h of the experiment. However, a significant shift in the conductance peak was observed, occurring at 20–22 h of the experiment and reaching a value of 900 nS (Figure 3). A delay in adhesion and EPS matrix formation was noted until the 18th hour of the experiment. To assess the statistical significance of changes in the course of dynamics, an analysis of variance (ANOVA) with Fisher’s post hoc testing was performed (Figure 3b). The conductance changes in the PAO1 biofilm in the presence of KP15, KP34, KP36, and KTN6 phages were compared, along with the variations between the tested phage options. The dynamic of the *P. aeruginosa* PAO1 biofilm was significantly accelerated by phage KP34, as confirmed by statistical tests. No significant differences were observed in the dynamics of biofilm formation following the application of Klebsiella phages KP15 and KP36 compared to the control culture. Similarly, no significant changes were observed with Pseudomonas phage KTN6, indicating that the nature of the biofilm and overall biofilm mass remained unchanged, even though the phage effectively delayed biofilm matrix formation.

### 3.2. No Effect of Non-Specific Klebsiella Phages on P. aeruginosa PAO1 Planktonic Culture

First, to confirm that selected Klebsiella phages do not specifically recognize PAO1 cell surface receptors in contrast to the Pseudomonas phage, an adsorption assay experiment was performed. Indeed, no reduction in free phage particles was observed for KP34 and KP36 Klebsiella phages compared to the 99% reduction for KTN6 phage after 12 min of bacteria–phage co-incubation (Appendix A).

In subsequent experiments, we investigated the impact of non-specific phages on the planktonic PAO1 population by analyzing growth dynamics using standard culture density and the resistance parameter (impedance) measurements (Figure 4). The resistance R100k_norm parameter drop allowed for the detection of changes in planktonic cell numbers and medium composition [31]. After the stabilization of the culture conditions, which lasted about 4 h from the start of the experiment, a gradual decrease from 0.97 to 0.85–0.80 in the R100k_norm parameter was observed for Klebsiella phages KP15, KP34, and KP36, similar to the PAO1 control (Figure 4a). This result indicates that bacteria were able to propagate regardless of non-specific phage presence. The findings were further confirmed in the OD_600_ experiments, where no differences were observed among the aforementioned samples and the control. A different trend was demonstrated in cultures infected with the specific phage Pseudomonas KTN6. For the first 14 h of the experiment, the R100k_norm dynamic resembled those of the negative medium control (TSB). However, after that time, the R100k_norm parameter decreased to 0.85, similar to the untreated control. This suggests that the KTN6 phage successfully inhibited the population growth initially, but phage-resistant clones began to rebuild the planktonic population after approximately 14 h, which was also confirmed by the phage susceptibility testing using the double-layer method (Appendix A). Similar trends were noted in the OD_600_ experiments (Figure 4c). Both measurement methods (impedance and optical density) demonstrated consistency with the behavior of PAO1 biofilm formation, as measured using the impedance. To evaluate the significance of observed changes in planktonic culture, statistical tests were carried out for the 4 h–24 h timeframe of the experiment (Figure 4b). The timeframe was selected based on the monitored changes in the G100m dynamics and the corresponding experiment that tracked changes in the optical density of the culture (Figure 4c). Statistical differences were observed only for the KTN6 phage treatment when compared to other tests and the growth control, demonstrating that Klebsiella phages were unable to effectively propagate on the PAO1 planktonic population. In contrast, the specific Pseudomonas lytic KTN6 phage significantly reduced the planktonic population dynamics.

### 3.3. No Effect of K. pneumoniae Culture Compounds on P. aeruginosa PAO1 Biofilm Dynamics

To confirm that the observed effects in PAO1 culture dynamics, as measured by impedance spectra, were caused by non-specific phage activity rather than intracellular components released during bacterial lysis, we performed experiments using bacterial filtrates. These filtrates were derived from the growth of uninfected bacteria under the same culture conditions used for phage propagation. Specifically, we aimed to determine whether components released during *Klebsiella* cell lysis could influence the biofilm formation kinetics or planktonic population dynamics of *P. aeruginosa* PAO1.

Changes in the G100m parameter were monitored for the filtrates, including Kp77, Kp486, and KpATCC strains, incubated with the *P. aeruginosa* PAO1 culture (Figure 5). The recorded changes in the G100m parameter for each test filtrate were very similar. After an initial stabilization period of approximately 4 h, the conductance reached about 300 nS and remained stable until approximately the 10th hour of the experiment. Then, a three-fold increase in electrical conductivity was observed across all experiments, reaching its maximum between the 12th and 14th hour at about 1 µS. This was followed by a two-fold decrease in the G100m parameter, starting around the 18th hour, which stabilized until the end of the experiments. Blank filtrate samples (filtrate without PAO1 culture, dash curves) showed trends similar to the negative control (medium only) (Figure 5a). These results proved that *K. pneumoniae* filtrates did not influence the growth of the PAO1 biofilm.

The high-frequency resistance (R100k_norm) of *P. aeruginosa* PAO1 cells incubated with *K. pneumoniae* culture filtrates showed the same change dynamics as the control culture of PAO1, thus confirming no effect on the growth dynamics of planktonic forms of *P. aeruginosa* PAO1 (Figure 5b). The negative control consisting of the filtrates alone (without *P. aeruginosa* PAO1 cells in the measurement wells) excluded any influence of the filtrates on the R100k_norm parameter during the 48 h experiment, as the blank curves closely resembled the TSA medium curve. In summary, these results proved that *K. pneumoniae* culture components released during cell lysis did not affect the biofilm formation kinetics and the planktonic population of *P. aeruginosa* PAO1. Therefore, the observed acceleration of PAO1 biofilm formation was attributable to the presence of the non-specific Klebsiella phage rather than any residual components from the bacterial host filtrates.

Given that phage infections activate cellular stress responses, potentially causing significant differences in the intracellular composition of lysates from phage-infected versus uninfected cells, we also tested phage lysates filtered through membranes with different molecular weight cutoffs (30 kDa and 10 kDa). After confirming that these lysates were free of phages, we assessed their impact on biofilm production.

Biofilm dynamics were recorded through simplified impedance measurement, with changes in the G100m parameter used to track biofilm development. Changes in planktonic forms were recorded in parallel. Regardless of the fraction used (10 kDa or 30 kDa), the culture dynamics followed a similar trajectory, with no statistically significant difference compared to the control biofilm culture in the TSB medium. Despite the observed shifts presented in Figure 6, statistical analyses confirmed that these differences were not significant. The G100m parameter stabilized during the first 4 h of the experiment. Around the 12th hour, the curves for the tested lysate fractions (KP15, KP34, KP36, and KTN6 phages represented by red, blue, yellow, and brown curves in Figure 6a,b, respectively) became steeper and reached a maximum in low-frequency conductance. Less than two hours later, an increase in the control biofilm was observed (green curve in Figure 6a,b). As indicated by statistical analyses (Figure 6c,d), these differences were not statistically significant. Regardless of the applied fraction (30 kDa or 10 kDa), the dynamics of biofilm formation by *P. aeruginosa* PAO1 cells were very similar. Therefore, the removal of phages and size-specific compounds released from both lysed cells and damaged virions did not affect PAO1 biofilm formation.

The growth dynamics of planktonic forms, measured using the R100k_norm parameter after the application of the 10 kDa and 30 kDa fractions of the tested lysates, show quite analogous behavior. Regardless of the fraction used, a quite clear decrease in the R100k_norm parameter compared to the TSB control was observed between the 4th and 12th hour of culture (Figure 7). In the following hours of the experiment, a slow decrease was recorded, with flattening of the curves for all options and reaching a value of about 0.85. The dynamics during the first 12 h were similar across all conditions, indicating no significant influence (stimulation or inhibition) of the phage lysate fractions on PAO1 cell division. After 12 h, the PAO1 control (green curve in Figure 7a,b) slightly diverged from the others. However, the difference in these dynamics was not significant, as indicated by statistical analyses.

## 4. Discussion

Due to the complexity of its spatial structure, ability to communicate at the molecular level, and adaptation to changing environmental conditions (e.g., horizontal gene transfer), bacterial biofilms seem to be a bastion of microorganisms that are difficult to eradicate. The features characterizing biofilms constitute a serious problem, ranging from medical and industrial environments to adverse effects on the ecosystem [34]. It is, therefore, crucial to monitor and control the level of this threat. The increasing bacterial drug tolerance or resistance to various antibiotics has highlighted the need for alternative therapies. This has led to the use of natural bacterial enemies—bacteriophages—as a means to control both planktonic and biofilm-forming bacteria [12]. The selection and optimization of phage therapies seemed to be the gold standard in neutralizing planktonic forms of bacteria. However, over millions of years, bacterial cells have co-evolved with their natural predators, often forming mutualistic systems that have been beneficial for both parties throughout evolution [35]. Despite this, bacteria developed various strategies to resist viral infection, being a stress factor for the cell population [36]. One of the most physiologically and ecologically complex ways this occurs is the modification of phage receptors or the production of biofilm structures, which allow susceptible bacteria to be hidden within the EPS matrix [34]. Throughout evolution, bacteriophages found a way to reach sessile cells through the production of numerous enzymes, such as polysaccharide depolymerases, lipases, and DNA-azes, facilitating the cleavage of the biofilm matrix [37]. One example is the depolymerases depoKP34 and depoKP36 derived from Klebsiella KP34 and KP36 phages degrading the *K. pneumoniae* capsule and EPS [16,28].

The phage impact on bacterial biofilm formation was reported by other researchers in studies focusing on the phage dose adjustment of polyvalent PEB phages against the mixed *P. aeruginosa* and *E. coli* biofilm [17]. These experiments demonstrated the effect of hormesis, in which a two-phased dose–response relationship was observed while optimizing the most efficient anti-biofilm phage titer. It was found that the synthesis of exopolysaccharides and eDNA increased at lower VBR. At the same time, the expression of QS genes was triggered, and a more structured phage-resistant biofilm was formed [17]. A *Vibrio cholerae* study proved that bacteria are able to increase the biofilm formation in response to *V. cholerae* cellular components released through lysis (norspermidine), regardless of the nature of the agent causing the lysis, for instance, the lytic phage [38]. In contrast, *P. aeruginosa* tested in our experiments did not sense the mechanically produced *K. pneumoniae* lysate, probably because such sensing is species/genus specific.

It should be emphasized here that, unlike our group, the aforementioned studies reported the hormetic effect for phages specific to the bacterial host.

There are indications that phages may utilize bacterial communication systems based on QS. The effect on extracellular communication could potentially inactivate biofilm-forming cells. Phages possess genes for the receptors that bind to autoinducers originally expressed on bacterial cells. When the phage receptor binds to the host-produced autoinducer, it triggers the phage lysis program. Phages can communicate and control their lytic/lysogenic cycle switch via the arbitrium molecules, which count the viral particles similarly to the QS mechanism used by bacteria to monitor cell density [39].

Nevertheless, there is not much known about the phage–bacteria interactions between non-infecting viruses and unsusceptible hosts, to understand the ecological and evolutionary dynamics between phages and bacteria in the environmental niche. The phage–host interplay is usually studied regarding physical interactions between tailed phages and their prey (bacterial pathogens), accomplished by virion-associated receptor-binding proteins (tail spikes or tail fibers) and bacterial cell surface macromolecules (LPS, OMP, flagella, capsule, etc.) [40].

The development of new research techniques, alongside the continued use of classical methods, enables both the direct and indirect monitoring of biofilm biology, including phage–bacteria interactions. The measurement technique proposed by our team, based on impedance spectroscopy, is a non-invasive tool that does not require sample manipulation and enables the real-time monitoring of biofilm dynamics. The results obtained by our group in previous reports using impedance techniques correlated well with those from spectrophotometric methods, microscopy, or classical colony counting [30,41].

In the presented study, we aimed to investigate the non-specific interactions between phages and bacteria using the *P. aeruginosa* PAO1 strain model, along with lytic phages specific to *K. pneumoniae.* Our objective was to determine whether bacteria could sense the presence of phage particles and initiate the innate mechanisms of cell protection against potential viral infections. If the assumption holds, we should observe the effect of non-specific/non-propagating phages on the model bacterial population. By combining impedance and standard optical density techniques, we were able to monitor the influence of phages on the biofilm and planktonic population dynamics of the *P. aeruginosa* PAO1 strain. For the impedance measurement, two electrical parameters, G100m and R100k_norm, were monitored using simplified analysis (transformations and modeling in [30] and shortly described in Appendix A). The first parameter characterizes near-electrode phenomena, such as adhesion, maturation, matrix EPS remodeling, and dispersion. The high-frequency resistance (R100k_norm) provides an image of changes in the planktonic culture. By analyzing the course of both parameters, we observed a biofilm acceleration of approximately 2 h for non-specific phages, especially for podovirus KP34, compared to the PAO1 control that was not treated with phages (Figure 3). In contrast, the curves of the R100k_norm parameter show no effect of Klebsiella phages on planktonic PAO1 culture (Figure 4).

The unspecific reaction of PAO1 culture to the presence of Klebsiella phage lysate could draw an alternative hypothesis regarding the effect of the cell debris and/or membrane vesicles formed during the phage-mediated lysis. Assuming that the features of vesicles formed due to the spanin-mediated fusion of the internal and outer membranes may be substantially different from the vesicles formed during the UV disintegration of the *K. pneumoniae* cells (our negative control). Nevertheless, such a hypothesis is rather unlikely based on the *V. cholerae* report, proving that mechanically released cellular components (norspermidine) without lytic phage presence induce biofilm formation in vibrio populations [38]. Secondly, our experiments showed that only one *K. pneumoniae* lysate (filtrate of podovirus KP34) impacted *P. aeruginosa* biofilm formation. Moreover, the siphovirus KP36 was propagated on the host (Kp77), differing only in plasmid drug resistance genes compared to Kp486—the host of phage KP34. Thus, Kp77 and Kp486 phage-mediated debris did not differ significantly in composition [42].

It is also worth emphasizing that the Klebsiella phage virions used in our model were not considered ligands to *Pseudomonas* cellular receptor(s); thus, none of the specific interactions were expected. Moreover, Klebsiella phage KP34 and KP36 possess the RBP, recognizing the same receptor—capsule of the K63 serotype, which does not exist in *P. aeruginosa*. Of course, the elucidation of the phenomena we observed should be further investigated using other techniques, including molecular, physical, and genetic.

In summary, our study provides new insight into non-specific phage–bacterial interactions, which, as we hypothesize, are likely based on the mechanistic sensing of phage virion presence in the bacterial surroundings. In response to the potential viral infection, this sensing triggers the formation of the protective biofilm matrix to block probable phage adsorption and inhibit the initial step of phage propagation. However, this far-reaching hypothesis requires further verification using other techniques that can offer a deeper understanding of the ecology and co-evolution between phages and their hosts. Advances in this research area could be crucial for improving our understanding of the phage impact on bacterial populations and may guide the development of new antibiofilm therapies.

## Figures and Tables

**Figure 1 viruses-17-00615-f001:**
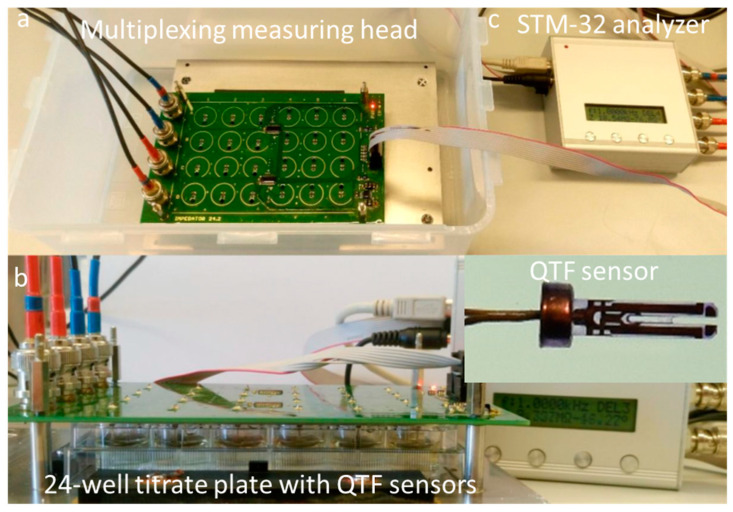
Parts of the measurement setup, including multiplexing measuring head (**a**), 24-well titration plate with QTF sensors (**b**), and the STM-32 analyzer (**c**) [30].

**Figure 2 viruses-17-00615-f002:**
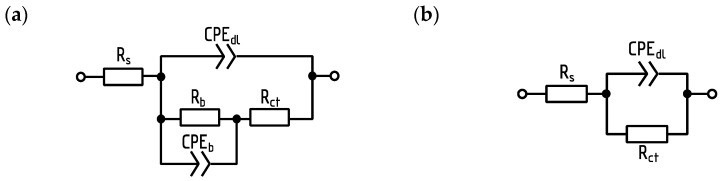
Primary (**a**) and simplified (**b**) electric equivalent circuits (EECs) used for the analysis of the measured impedance spectra [30].

**Figure 3 viruses-17-00615-f003:**
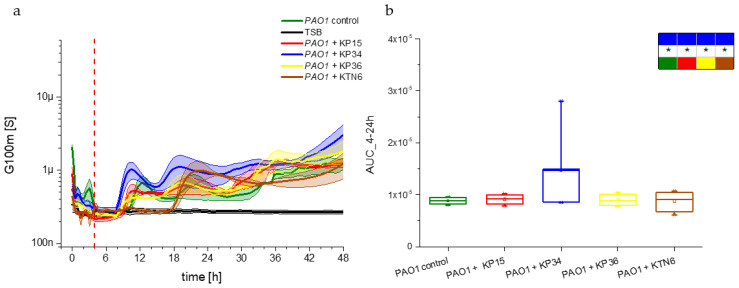
(**a**) Biofilm growth dynamics measured by the G100m parameter for *P. aeruginosa* PAO1 culture. The positive control consisted of untreated culture (green curve), negative control of TSA medium (black curve), and test samples treated with phages at VBR = 1 with specific Pseudomonas phage KTN6 (brown curve), and Klebsiella phages KP15 (red curve), KP34 (blue curve), KP36 (yellow curve). (**b**) Variability of the G100m parameter for the area under the curves estimated for the range from 4 to 24 h of the experiment. The boxes present the statistical significance analysis with the colors corresponding to the panel (**a**). The asterisk indicates a significant level of *p* > 0.05.

**Figure 4 viruses-17-00615-f004:**
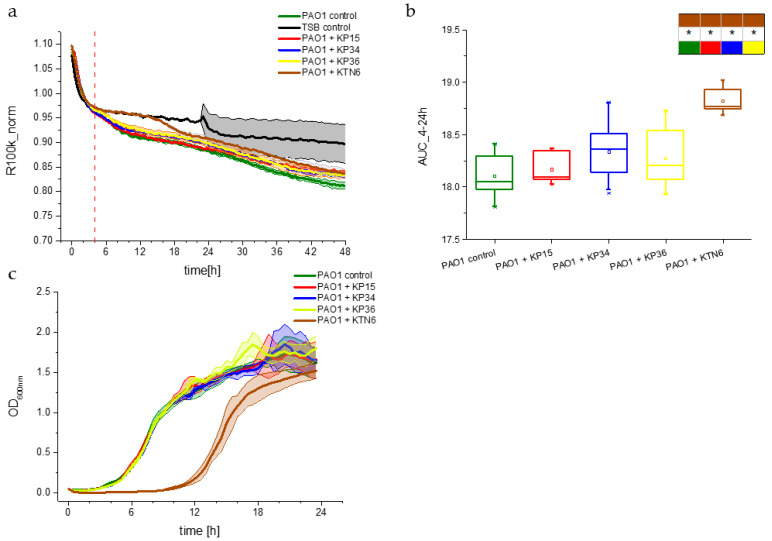
(**a**) Planktonic growth dynamic measured by the R100k_norm parameter for *P. aeruginosa* PAO1. The positive control consisted of untreated culture (green curve), negative control of TSA medium (black curve), and test samples of treated with phages at VBR = 1 with specific Pseudomonas phage KTN6 (brown curve), and Klebsiella phages KP15 (red curve), KP34 (blue curve), KP36 (yellow curve). (**b**) Box chart of the variability of the R100k_norm parameter for the area under the curves estimated for the range from 4 to 24 h of the experiment. The boxes present the statistical significance analysis with the colors corresponding to the panel (**a**). The asterisk indicates a significance level of *p* > 0.05; (**c**) OD_600_ measurement of planktonic PAO1 in the presence of phages (VBR = 1).

**Figure 5 viruses-17-00615-f005:**
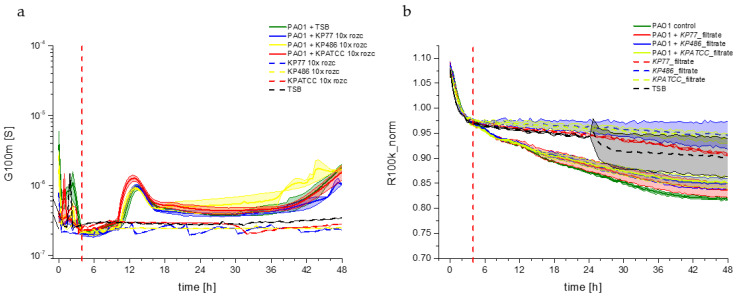
(**a**) Biofilm growth dynamics of impedance measured by the G100m parameter and (**b**) Resistance parameter R100k_norm of planktonic forms of *P. aeruginosa* PAO1 as a control (green curve) and in the presence of Kp77 (blue curve) and Kp486 (yellow curve), KpATCC (red curve) culture filtrates, and the appropriate blank filtrate options (dash curves).

**Figure 6 viruses-17-00615-f006:**
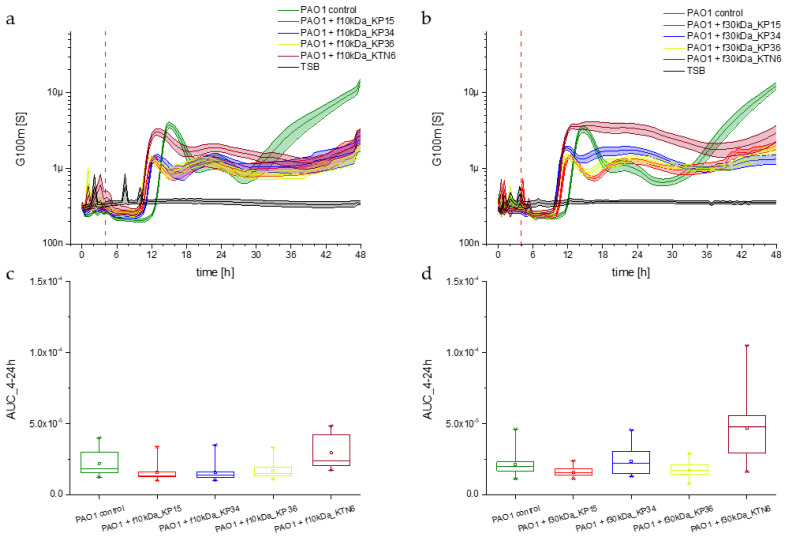
(**a**) Biofilm growth dynamics after application of 10 kDa fraction of phage lysates KP15 (red curve), KP34 (blue curve), KP36 (yellow curve) KTN6 (brown curve) to *P. aeruginosa* PAO1 (green curve) culture measured by impedance the G100m parameter. (**b**) Impedance measurement with detection of the G100m parameter of PAO1 cultures with phage lysates KP15 (red curve), KP34 (blue curve), KP36 (yellow curve) KTN6 (brown curve) fractionated with a 30 kDa membrane. Statistical calculations for the tested 10 kDa and 30 kDa filtrate options significance level of *p* > 0.05 ((**c**,**d**), respectively).

**Figure 7 viruses-17-00615-f007:**
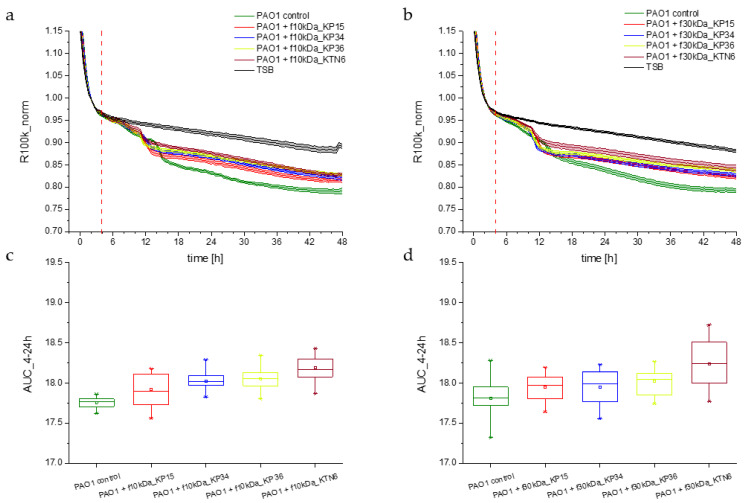
(**a**) Planktonic growth dynamics after application of 10 kDa fraction of phage lysates KP15 (red curve), KP34 (blue curve), KP36 (yellow curve) KTN6 (brown curve) to *P. aeruginosa* PAO1 (green curve) culture measured by impedance the R100k_norm parameter. (**b**) Impedance measurement with detection of the R100k_norm parameter of PAO1 cultures with phage lysates KP15 (red curve), KP34 (blue curve), KP36 (yellow curve) KTN6 (brown curve) fractionated with a 30 kDa membrane. Statistical calculations for the tested 10 kDa and 30 kDa filtrate options significance level of *p* > 0.05 ((**c**,**d**), respectively).

**Table 1 viruses-17-00615-t001:** The collection of bacteriophages used in this work.

Phage	Taxonomy	Bacterial Host	Characteristics	Reference
KP15	*Slopekvirus*	*K. pneumoniae* ATCC 700601	myovirus targeting proteinous receptor	[26]
KP34	*Drulisvirus*	*K. pneumoniae* 77	podovirus producing K63 capsule-degrading depolymerase	[26,27]
KP36	*Webervirus*	*K. pneumoniae* 486	siphovirus producing K63 capsule-degrading depolymerase	[26,28]
KTN6	*Pbunavirus*	*P. aeruginosa* PAO1	myovirus targeting LPS	[29]

## Data Availability

The original contributions presented in this study are included in the article/Appendix A. Further inquiries can be directed to the corresponding author.

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
