# Peer review of "Klebsiella Lytic Phages Induce Pseudomonas aeruginosa PAO1 Biofilm Formation"

_viruses, 2025, doi:10.3390/v17050615_

Round 1

Reviewer 1 Report (New Reviewer)

Comments and Suggestions for Authors

Comments to the Author

This manuscript reports on viral-host interactions in order to determine if bacteria can detect phage particles, regardless of specificity, and triggering protective mechanisms such as biofilm matrix formation to block potential phage adsorption and infection. Biofilm formation was evaluated by using impedance and optical density for planktonic cells.

M&M section

Please explain more clearly the relation between the variables evaluated in relation with the biofilm formation.

Results section

The G100m parameter stabilized during the first 4th hour of the experiment”: what append during this 4-h period?

Taking into account the variability observed in Figure 3, it seems that each strain-phage pair generates a specific graphic, like a fingerprint, since they are different from each other. How different are the phages tested in order to generate these curves? How reproducible are the results obtained with this technique?

Furthermore, after 48 h of incubation it can be observed that most curves continue to increase: Biofilms induction, i.e., analyze the beginning, however, have you carried out tests that extend to incubation times greater than 48 h?... taking into account that biofilms take time to develop.

On the other hand, why were these results not compared with a technique such as that which uses crystal violet to measure biofilms formation? This technique is very simple and can provide important information to corroborate the results found in this work.

Discussion section

Lines 426 to 432: several references are missing.

In the Discussion section the authors say that “The results obtained using this technique correlate well with those 451 from spectrophotometric methods, microscopy, or classical colony counting”, however, in the present work such comparisons were not made.

Lines 460 to 467 belongs to the M&M section.

The discussion section is written in a confusing way, it only talks about hypotheses without the support of the results obtained in the present work. Experiments should be discussed in relation to the results found by other authors. Furthermore, the manuscript does not highlight the novelty of the work.

Minor comments

Please replace hours with h through the manuscript.

Author Response

Reviewer 1

M&M section

Q1: Please explain more clearly the relation between the variables evaluated in relation with the biofilm formation.

A1: We are very grateful for drawing attention to the essence of our measurements. We have been conducting bioimpedance measurements for microbiological applications since about 2011. We were the first to conduct long-term experiments (even up to 8 days of biofilm cultivation) using QTF-type sensors [Piasecki et al., 2013;doi.org/10.1016/j.snb.2012.12.087]. Due to the measurement range and the analysers used at that time, we were able to observe biofilm dispersion processes in bacterial cultures. Extending the precision of the measurement with a dedicated and specially built analyser allowed for observations of biofilm adhesion, maturation, and, in the later hours, dispersion. During the work, analyses of impedance spectra were conducted, which is de facto the first result obtained from the sensor. Impedance spectra, the actual changes recorded by sensors, became the basis for modeling using equivalent electrical models (this is a natural process also determined during corrosion of metal surfaces). Based on the models, changes can be visualized using certain parameters. In the case of working with biofilms, two of them correspond well with biological measurements. The Rs parameter measured at a frequency of 100kHz allows us to determine the growth dynamics of planktonic forms of bacteria in the experiments. We defined it as R100k_norm, because each sensor may have minor post-production defects, therefore in biological experiments, obtaining information from several sensors, we normalized this value. Nevertheless, it corresponds well to the CFU measurement. The second parameter, Qb (from biofilm), derived from the analysis of the EEC model, is related to conductance at a lower frequency of 100mHz, therefore, we defined it as G100m. This parameter concerns biological phenomena occurring on the sensor surface. These changes indicate cell adhesion to the sensor, biofilm maturation, and its dispersion (depending on the timeline, of course). We showed the first stages, i.e. adhesion and maturation, in experiments conducted for 48 hours. These parameters were described quite briefly because we tried not to duplicate information from a more detailed analysis, which we presented in an earlier publication [Guła et al., 2020; doi:10.3390/v12040407].

We have added the Introduction to impedance spectroscopy measurements of bacterial biofilms using the QTF platform based on our previous reports Piasecki et al., 2013; doi.org/10.1016/j.snb.2012.12.087, and Guła et al., Viruses, 12(4), 407. doi.org/10.3390/v12040407 to the Supplementary Materials.

Results section

Q2:The G100m parameter stabilized during the first 4th hour of the experiment”: what happened during this 4-h period?

A2: Thank you very much for drawing attention to this aspect. During sensitive measurements of G100m conductivity associated with biofilm, the first 4 hours of detection are associated with standardization of measurement conditions. The G100m parameter can be influenced by the medium environment, which stabilizes at a new temperature (transition from room temperature to 37°C), which involves changes in the density and viscosity of the medium (which is influenced by the nutrients of the medium), as a result of which it affects the adaptation of cells to new conditions and may result in the secretion of substances by cells that facilitate adhesion to the sensor surface. All these elements can interfere with conductivity measurements in the first 4 hours of the experiment. Even in the measurements of the medium control or bacteriophage control, some parameter noise is observed at this early stage of the experiments, which confirms the assumption about the stabilization of the temperature and the viscosity and density of the culture medium.

Q3: Taking into account the variability observed in Figure 3, it seems that each strain-phage pair generates a specific graphic, like a fingerprint, since they are different from each other. How different are the phages tested in order to generate these curves? How reproducible are the results obtained with this technique?

A3: Thank you very much for this question. It is very important for the proposed technique and the expansion of our measurement system with further biological experiments. The measurement fidelity and repeatability are very high. In all the experiments with this technique so far, we have used hundreds, maybe even thousands of disposable QTF type sensors. In our two earlier publications, we assumed three biological repetitions of the experiments during the measurements, each performed in four technical repetitions. For the PAO1 + phage KP34 experiment, we previously performed three additional biological repetitions. Based on the results and our impedance spectroscopy experience, we believe that this is a highly reproducible technique.

We selected four phages differing at the genomic and protein level, belonging to three main virion morphotypes, which influence also the mechanism of host recognition and DNA ejection. Nevertheless, the fingerprints of phage-host interaction are a common feature, even if we compare the infection of one specific phage in two different hosts.

Q4: Furthermore, after 48 h of incubation, it can be observed that most curves continue to increase: Biofilms induction, i.e., analyze the beginning, however, have you carried out tests that extend to incubation times greater than 48 h?... taking into account that biofilms take time to develop.

A4: In the present work, we focused on two stages of the biofilm cycle. We wanted to monitor adhesion and the initial stages of biofilm maturation, hence the visible increases in Figure 3 of the G100m parameters. In screening experiments, we performed a 5-day incubation (in earlier experiments we extended the measurements to even 10 days). As we explained the dynamics of changes in biofilm growth in our previous reports, conducted experiments in a ten-day setup [Guła et al., 2020; doi:10.3390/v12040407]. We observed the behavior of biofilm culture based on two phases: maturation and dispersion. In the current publication, we focused attention on the initial stages of biofilm formation. We were interested in changes in the initial hours of stationary culture. Due to the many controls, tests with bacterial lysates, the controls with phages alone, and fractions after lysed phages, experiments longer than 48 hours would be very time-consuming. Preparing sensors and conducting measurements would result in a significant extension of the experimental time, and this was very important to us - to obtain repeatable results and monitor earlier phases in the biofilm formation cycle. This does not change the fact that we can conduct such analyses for much longer than 48 hours.

Q5: On the other hand, why were these results not compared with a technique such as that which uses crystal violet to measure biofilm formation? This technique is very simple and can provide important information to corroborate the results found in this work.

A5: Thank you for this remark. We would like to explain a little about this matter. When we started working with QTF-type sensors as a new measurement technique, we performed a series of comparable experiments, including the use of classic types of biological experiments:  CV, CFU, OD600nm, visualizations using light and electron microscopes. Crystal violet served as a validator of the validity of using the mentioned sensors in biofilm studies, which we presented in a publication from 2013 (Figure 6) [Piasecki et al., 2013;doi.org/10.1016/j.snb.2012.12.087]. It was this publication that inspired us to conduct further experiments monitoring changes in biofilm growth in its first 48 hours. The difference is that in the present work, we used a more precise and, therefore, sensitive impedance analyzer with more precise spectrum analysis algorithms, which effectively monitored electrical changes already in the initial stages of the experiments. Based on the SEM technique or CFU measurements in the 2020 publication [GuÅ‚a et al., 2020; doi:10.3390/v12040407], we considered that these previous data would be sufficient as a form of control for the measurement using the impedance spectroscopy technique. The assumptions of the current publication included many controls, and a large group of tests with phage lysates and filtrates, which is why we considered that it would be time-consuming to add measurements using classical methods. Here, attention must be paid to the measurement resolutions obtained by different methods. The impedance spectroscopy technique provides information in real-time every 25-30 minutes for one sensor. The most important, also statistically significant changes in dynamics occur between the 6th and 24th hour of bacterial incubation. It would be difficult to carry out measurements with crystal violet at a similar point resolution.

Discussion section

Q6: Lines 426 to 432: several references are missing.

A6: Missing references have been added.

Q7: In the Discussion section the authors say that “The results obtained using this technique correlate well with those from spectrophotometric methods, microscopy, or classical colony counting”, however, in the present work such comparisons were not made.

A7: In this shortcut, we meant our first experiences with the use of QTF as biofilm detection sensors (not only as impedance sensors). Appropriate corrections have been made.

Q8: Lines 460 to 467 belong to the M&M section.

A8: The lines have been moved to the M&M section.

Q9: The discussion section is written in a confusing way, it only talks about hypotheses without the support of the results obtained in the present work. Experiments should be discussed in relation to the results found by other authors. Furthermore, the manuscript does not highlight the novelty of the work.

A9: We thank the reviewer for his/her invaluable comments, which helped us significantly improve our manuscript. We have expanded the discussion better to explain our findings in the context of existing literature.

Minor comments

Q10: Please replace hours with h through the manuscript.

A10: The text has been corrected according to the suggestion.

Reviewer 2 Report (New Reviewer)

Comments and Suggestions for Authors

The manuscript titled  “Klebsiella lytic phages induce Pseudomonas aeruginosa PAO1  biofilm formation” is devoted to investigation of viral-host interactions between non-host-specific phages and Pseudomonas aeruginosa, assessing whether bacteria can detect phage particles and initiate protective mechanisms. Using real-time biofilm monitoring by impedance and optical density techniques, Authors have monitored phage effects on biofilm and planktonic populations. Three Klebsiella phages of different taxonomic groups —Slopekvirus, Drulisvirus, and Webervirus were tested against P.  aeruginosa PAO1 population, as well as Pseudomonas-specific Pbunavirus.

The obtained results indicated that Klebsiella phages non-specific to P. aeruginosa, particularly podovirus KP34, accelerated biofilm formation without affecting planktonic cultures. Authors proposed the hypothesis suggesting that bacteria possess specific mechanism to sense phage virions, regardless of specificity, triggering biofilm matrix formation to block potential phage adsorption and infection.

Nevertheless, further research is needed to understand the ecological and  evolutionary dynamics between phages and bacteria.

The manuscript is well-written, and both applied method and results are justified by previous publications including careful description done in previous publications including work of GuÅ‚a, G., Szymanowska, P., Piasecki, T., Góras, S., Gotszalk, T., & Drulis-Kawa, Z. (2020). The Application of Impedance Spectroscopy for Pseudomonas Biofilm Monitoring during Phage Infection. Viruses12(4), 407. https://doi.org/10.3390/v12040407

One remark on repeated use on Page 4  Figures 1 and 2 ( Parts of the measurement setup, including multiplexing measuring head and Primary (a) and simplified (b) electric equivalent circuits (EECs) used for the analysis of  the measured impedance spectra). They repeat in general figures published in above mentioned work of G. GuÅ‚a et al (2020). It is reasonable to move them back to the Supplement.

One more question for the statement at Line 307:

“…the trend demonstrated in cultures infected with the specific phage Pseudomonas KTN6. For the first 14 hours of the experiment, the R100k_norm dynamic resembled those of the negative medium control 309 (TSB) However, after that time the R100k_norm parameter decreased to 0.85 value, similar to the untreated control. This suggests that KTN6 phage successfully inhibited the  population growth initially, but phage-resistant clones began to rebuild the planktonic population after approximately 14 hours. Similar trends were noted in the OD600 313 experiments (Figure 4c).”

The question: Have you validated the presence of phage-resistant clones in this or similar variants in other experiments, or is this a pure speculation?  

Author Response

Reviewer 2

The manuscript titled  “Klebsiella lytic phages induce Pseudomonas aeruginosa PAO1  biofilm formation” is devoted to investigation of viral-host interactions between non-host-specific phages and Pseudomonas aeruginosa, assessing whether bacteria can detect phage particles and initiate protective mechanisms. Using real-time biofilm monitoring by impedance and optical density techniques, Authors have monitored phage effects on biofilm and planktonic populations. Three Klebsiella phages of different taxonomic groups —Slopekvirus, Drulisvirus, and Webervirus were tested against P.  aeruginosa PAO1 population, as well as Pseudomonas-specific Pbunavirus.

The obtained results indicated that Klebsiella phages non-specific to P. aeruginosa, particularly podovirus KP34, accelerated biofilm formation without affecting planktonic cultures. Authors proposed the hypothesis suggesting that bacteria possess a specific mechanism to sense phage virions, regardless of specificity, triggering biofilm matrix formation to block potential phage adsorption and infection.

Q1: Nevertheless, further research is needed to understand the ecological and evolutionary dynamics between phages and bacteria.

A1: Indeed, we observed a certain phenomenon using a measurement technique enabling the detection of subtle interaction between phages and a non-specific host. This is of course, only the tip of the iceberg for further experiments in the ecology and evolution of phage-bacteria dynamics, which should be investigated at various levels in future studies.

Q2: The manuscript is well-written, and both applied method and results are justified by previous publications including careful description done in previous publications, including work of GuÅ‚a, G., Szymanowska, P., Piasecki, T., Góras, S., Gotszalk, T., & Drulis-Kawa, Z. (2020). The Application of Impedance Spectroscopy for Pseudomonas Biofilm Monitoring during Phage Infection. Viruses12(4), 407. https://doi.org/10.3390/v12040407. One remark on repeated use on Page 4  Figures 1 and 2 ( Parts of the measurement setup, including multiplexing measuring head and Primary (a) and simplified (b) electric equivalent circuits (EECs) used for the analysis of the measured impedance spectra). They repeat in general figures published in the above-mentioned work of G. GuÅ‚a et al (2020). It is reasonable to move them back to the Supplement.

A2: Thank you for this remark, nevertheless, all other reviewers suggested even enlarging this part of the text to make the reader more familiar with the impedance technique.

Q3: One more question for the statement at Line 307:“…the trend demonstrated in cultures infected with the specific phage Pseudomonas KTN6. For the first 14 hours of the experiment, the R100k_norm dynamic resembled that of the negative medium control (TSB). However, after that time, the R100k_norm parameter decreased to 0.85 value, similar to the untreated control. This suggests that KTN6 phage successfully inhibited the population growth initially, but phage-resistant clones began to rebuild the planktonic population after approximately 14 hours. Similar trends were noted in the OD600 313 experiments (Figure 4c).” The question: Have you validated the presence of phage-resistant clones in this or similar variants in other experiments, or is this a pure speculation?

A3: According to the reviewer's suggestion, we have added the experiment proving the presence of a phage-resistant population selected during phage infection (Supplementary Materials Fig.S5).

The general response to the reviewer: We would like to thank the reviewer for his/her invaluable comments, which helped us significantly improve our manuscript. We have also expanded the discussion better to explain our findings in the context of existing literature.

Reviewer 3 Report (New Reviewer)

Comments and Suggestions for Authors

The study by Cula et al. uses a novel method of impedance spectroscopy to measure the effect of the non-host bacteriophages on the Pseudomonas aeruginosa biofilm development. This method allows to follow the biofilm formation dynamics over the incubation time that is a very important improvement compared to classical biofilm end-point quantification with staining and elution. The application of this technique allowed the authors to demonstrate some increase in biofilm formation in presence of the phage lysate which disappeared  if the lysate components >30KDa were removed by filtration. The authors speculate that bacteria may somehow sense the presence of phage visions to alter their physiology.
Although the study is generally interesting and the finding strikes me as important, the data of the impedence spectrometry are hard to interpret from biological perspective. The authors operate the values such as G100m used as a proxy of biofilm abundance. However it is not clear for the reader how these are linked to the biofilm mass or cell number in the biofilm etc. The calibration of the method with conventional measurements in a subset of the samples in parallel to the impedance-based experiments would be very useful.
The conclusion is not well supported by the data. Although the results with the lysate fractioning are suggestive for the interpretation made by the authors, alternative hypotheses still cannot be excluded. For example, the cell debris and/or membrane vesicles formed during the phage mediated lysis could be responsible for the effect. It should be pointed out that the features of these vesicles formed due to the spanin-mediated fusion of the internal and outer membranes may be substation ally different from the vesicles formed during the UZ desintegration of the cells.
I'd suggest to prepare highly purified phage suspensions using gradient centrifugation or other methods and test if the pure visions can cause the same effect. Also the quantification of the phages based on biological titers is not enough if the phages are considered as a ligand to some hypothetical cellular receptor(s). The estimation of the physical concentration of phage particles would be more informative.

The description of the impedance spectrometry method and the device used is not enough for the reproduction of the experiments. I suggest deposition to Supplementary material of the full description of the method physics and detailed information sufficient for building the device (alternatively, if it is available commercially please provide the relevant information). At the same time the principles of the method in the main body of the paper may be compacted and simplified. 

The overall language and style of the manuscript are good although an additional round of polishing would be useful. Please give attention to proper introduction of the abbreviations when the are used for the first time.

Minor motes

Line 119. Probably the heads were submerged in isopropanol for 24 h rather rinsed by it for so long time
Line 127. Isopropanole was allowed to evaporate, not the sensors. The later were just dried
Line 129 What do you mean by the passivation?
Line 138 What is imaginary unit? A constant?
Lines 158-159  CPE has a capacity, but you introduce 2 different papameters, please define them
Line 183 MOI = multiplicity of infection or inoculation. Since the phages are not infecting in this setup I'd suggest to use different abbreviation, e/g. VBR (virus to bacteria ratio)
Line 248 replace "automatic" by "automated"
Line 276 The sentence is not clear, please rephrase

Author Response

Reviewer 3

The study by Cula et al. uses a novel method of impedance spectroscopy to measure the effect of the non-host bacteriophages on the Pseudomonas aeruginosa biofilm development. This method allows to follow the biofilm formation dynamics over the incubation time that is a very important improvement compared to classical biofilm end-point quantification with staining and elution. The application of this technique allowed the authors to demonstrate some increase in biofilm formation in presence of the phage lysate which disappeared  if the lysate components >30KDa were removed by filtration. The authors speculate that bacteria may somehow sense the presence of phage visions to alter their physiology.

Q1: Although the study is generally interesting and the finding strikes me as important, the data of the impedance spectrometry are hard to interpret from biological perspective.

A1: Indeed, the use of this impedance technique seems to make it difficult to assess biological phenomena. In our earlier publications, we tried to confirm the validity of using the new measurement tool in an accessible way and based on classical methods [Piasecki et al., 2013;doi.org/10.1016/j.snb.2012.12.087; Guła et al., 2020; doi:10.3390/v12040407]. The current results are based on many years of experience in impedance spectroscopy measurements and its application to monitor model bacterial biofilms. The impedance setup enables monitoring the dynamics of biofilm growth in real-time, in contrast to for instance crystal violet staining, which is also not specific (cells and matrix EPSs are stained). Using the impedance device we can measure, without the need for preparation, i.e. directly, changes in the dynamics of planktonic forms (R100k_norm measurement) and monitor the biofilm formation at each stage (G100m). This allows for the detection of dynamic changes with a resolution of 25-30 minutes. The 24 sensors working together significantly speed up obtaining repeatable results. Our previous reports compared the impedance results with classical techniques such as CV, OD600, CFU, and microscopic visualizations, and we could successfully detect all stages of biofilm formation.

We have added the Introduction to impedance spectroscopy measurements of bacterial biofilms using the QTF platform based on previous reports Piasecki et al., 2013; doi.org/10.1016/j.snb.2012.12.087, and Guła et al., Viruses, 12(4), 407. doi.org/10.3390/v12040407 to the Supplementary Materials.

Q2: The authors operate the values such as G100m used as a proxy of biofilm abundance. However, it is not clear to the reader how these are linked to the biofilm mass or cell number in the biofilm etc.

A2: A valid issue has been raised here, for which we are very grateful. In our work, we operate on certain parameters that result from earlier observations (those published earlier). We did not want to duplicate data from previous publications in this work, which is why the presented processes were described in a minimalist form. Our technique is based on the detection of two changes in two parameters. One is related to the number of planktonic forms. We define this parameter as R100k_norm. With its help, we determine the influence of various factors (here bacteriophages or phage filtrates or bacterial host lysates) on the dynamics of cell count. The second parameter, somewhat similar to the crystal fillet, gives us information about changes in the biomass of the biofilm adhered to the surface of the QTF sensor. Similarly to using violet, we detect the formation, in the final stage, dispersion of the biofilm structure. We have a dynamic pattern for bacterial control, and against that, we monitor various changes and shifts in the time curve. This gives us a basis for applying hypotheses about the validity and effectiveness of various agents in the context of an untreated control biofilm. We can, therefore, reliably detect these differences and complement the panel of measurement tools already in use.

Q3: The calibration of the method with conventional measurements in a subset of the samples in parallel to the impedance-based experiments would be very useful.
The conclusion is not well supported by the data.

A3: This comment is extremely valuable for the application of our measurement system, and we are very grateful for it. We would like to provide some context regarding this issue. When we first began working with QTF-type sensors, which was a new measurement technique for us, we conducted a series of validation experiments, including classic biological experiments such as CV, CFU, OD600nm measurements, and visualizations using light and electron microscopes. Crystal violet was used as a validator to confirm the validity of using these sensors in biofilm studies, as detailed in our 2013 publication (Figure 6) [Piasecki et al., 2013; doi.org/10.1016/j.snb.2012.12.087]. The key difference in the present work is the application of a more precise and sensitive impedance analyzer, equipped with advanced spectrum analysis algorithms. This setup enabled us to monitor electrical changes in the early stages of the experiments. Based on our previous experiences, including data from SEM techniques and CFU measurements in our 2020 report, we considered the earlier data sufficient as controls for the impedance spectroscopy technique [Guła et al., 2020; doi:10.3390/v12040407]. Our current publication includes a large number of controls and tests with phage lysates and filtrates, which made it impractical to include additional measurements using classical methods. It is important to note the differences in measurement resolution between the various methods. The impedance spectroscopy technique provides real-time data every 25-30 minutes per sensor, and the most significant dynamic changes, both biologically and statistically, typically occur between the 6th and 24th hour of bacterial incubation. Carrying out measurements with crystal violet at these time points would have been highly time-consuming. Moreover, the limitation of the CV technique is the preparation step, giving mostly non-repeatable results for P. aeruginosa biofilm.

Q4: Although the results with the lysate fractioning are suggestive of the interpretation made by the authors, alternative hypotheses still cannot be excluded. For example, the cell debris and/or membrane vesicles formed during the phage-mediated lysis could be responsible for the effect. It should be pointed out that the features of these vesicles formed due to the spanin-mediated fusion of the internal and outer membranes may be substantially different from the vesicles formed during the UV desintegration of the cells.

A4: Of course, this aspect could also be investigated, complementing the current phenomenon. Nevertheless, this explanation is rather unlikely. Firstly, we separated debris and possible membrane vesicles by 0.22 µm filtration. Secondly, only one filtrate of podovirus KP34 impacted P. aeruginosa biofilm formation. The siphovirus KP36 was propagated on the host (Kp77) differing only in plasmid drug resistance genes compared to Kp486 – the host of phage KP34. Thus, we would not assume that Kp77 and Kp486 phage-mediated debris differ significantly in composition.

Q5: I'd suggest to prepare highly purified phage suspensions using gradient centrifugation or other methods and test if the pure visions can cause the same effect. Also the quantification of the phages based on biological titers is not enough if the phages are considered as a ligand to some hypothetical cellular receptor(s). The estimation of the physical concentration of phage particles would be more informative.

A5: In our study, Klebsiella phage virions were not considered ligands to Pseudomonas cellular receptor(s), thus, none of the specific interactions were expected. Moreover, phage KP34 and KP36 possess the receptor binding proteins (RBP) recognizing the same receptor – capsule of the K63 serotype, which does not exist in Pseudomonas aeruginosa. Of course, the elucidation of the phenomena should be further investigated by other techniques, including molecular, physical, and genetic.

Q6: The description of the impedance spectrometry method and the device used is not enough for the reproduction of the experiments. I suggest deposition to Supplementary material of the full description of the method physics and detailed information sufficient for building the device (alternatively, if it is available commercially please provide the relevant information). At the same time the principles of the method in the main body of the paper may be compacted and simplified.

A6:  Measurements by impedance spectroscopy are not a new phenomenon. Several works refer to this technique. Many of them are based on commercial impedance analyzers. In our initial experiences, we used analyzers such as Solartron 1260 (this is what we used in Piasecki et al., 2013; doi.org/10.1016/j.snb.2012.12.087 publication) or Keysight E4980A. Due to the dimensions of the devices and the huge costs of renting, the cooperating team built a compact analyzer IMP-STM32, which was characterized by a measurement range corresponding to the commercial Keysight E4980A. We used a dedicated impedance analyzer to obtain data for the 2020 publication [GuÅ‚a et al., 2020; doi:10.3390/v12040407]. It is worth noting a very important fact of our team's work. An important element of the measurement is the type of sensor. Commercially available finger electrode-type sensors or QCM-type microbalances can be used in the experiments. For our applications, a relatively cheap QTF tuning fork sensor platform was used. This is undoubtedly an original and unique solution used by our research team in other experiments. We have been successfully conducting our experiments with the QTF sensor platform for over a decade, achieving high repeatability of our measurement data.

According to the reviewer's suggestion, a full description of the method of physics and detailed information sufficient for building the device have been added to the supplementary materials.

The overall language and style of the manuscript are good although an additional round of polishing would be useful. Please give attention to proper introduction of the abbreviations when they are used for the first time.

Minor notes:

Q7: Line 119. Probably the heads were submerged in isopropanol for 24 h rather rinsed by it for so long time - corrected

Q8: Line 127. Isopropanole was allowed to evaporate, not the sensors. The later were just dried - corrected.

Q9: Line 129 What do you mean by the passivation?

A9: In the M&M section, we briefly explained the process of passivation of the metal parts of the sensor. Passivation is a general name for a process of thin layer formation that stops metal corrosion. The electrodes of QTFs are metal; due to our studies, mostly Al, which oxidize after removing the enclosure.

Q10: Line 138 What is the imaginary unit? A constant?

A10: We mean the imaginary unit in complex number. As is shown in line 136 the impedance is a value expressed with a complex number.

Q11: Lines 158-159  CPE has a capacity, but you introduce 2 different parameters, please define them

A11: The CPE is a generalised capacitor, and it has two parameters. The CPE would represent a regular capacitor if T=1, and then the value of Q would be related to the capacitance. If T <> 1, the Q does not directly represent the capacitance (its unit is S⋅s^T), but if T is close to 1, it may be approximated as such, but it is not strictly true. T, on the other hand, says how far the CPE is from a regular capacitor. Its values closer to 1 represent CPEs close to the capacitor, and the smaller the T further from the capacitor, the CPE is.

As such approximate explanations are not strictly true, we believe that they should not be included in the manuscript. We support the introduction of the CPE with reference to a handbook, one of several in which the CPE is described in detail.

Q12: Line 183 MOI = multiplicity of infection or inoculation. Since the phages are not infecting in this setup I'd suggest to use different abbreviation, e/g. VBR (virus to bacteria ratio) – corrected.

Q13: Line 248 replace "automatic" by "automated" – corrected.

Q14: Line 276 The sentence is not clear, please rephrase – rephrased.

The general response to the reviewer: We would like to thank the reviewer for his/her invaluable comments, which helped us significantly improve our manuscript. We have also expanded the discussion better to explain our findings in the context of existing literature.

Round 2

Reviewer 1 Report (New Reviewer)

Comments and Suggestions for Authors

The authors responded to reviewers' questions, and the manuscript was significantly improved. 

Author Response

Thank you for your positive review. We appreciate your thoughtful questions and are glad the manuscript has improved significantly.

Reviewer 3 Report (New Reviewer)

Comments and Suggestions for Authors

In the revised version the quality of the presentation of the background and data was significantly improved.  I think, however, that the authors should add more information on the last years data concerning phage stimulation of the biofilms especially the work appeared after the review [23] has been published. For example, two recent Vibrio cholerae papers strike me as highly relevant to this study (PMIDs 39753671, 38443393). At the same time both Introduction and Discussion sections may be compacted by using more concise style.

I do not agree with the authors' rebuttal of my request to perform additional controls with purified phage stocks. The conclusion made in this paper that phage particles themselves may trigger the bacterial response is very interesting and sounds groundbreaking. However, "the outstanding claims requires outstanding proof." The argument that all the vesicles are removed by 0.22 mkm filtration doesn't sound correct for me. Those having experience with phage microscopy know well that the vesicles are annoying contaminant in many phage preps even after gradient purification (the TEM control of the purified phage is necessary!). The size of (some fraction of) the vesicles is comparable to the phage virions. To summarize, I think that this paper should not be accepted until more direct proof of the phage particles involvement in the effect is provided. Without such a proof the data present have limited novelty since they demonstrate a previously described phenomenon by a relatively known but also already published method.

Author Response

Question 1: In the revised version the quality of the presentation of the background and data was significantly improved.  I think, however, that the authors should add more information on the last years data concerning phage stimulation of the biofilms especially the work appeared after the review [23] has been published. For example, two recent Vibrio cholerae papers strike me as highly relevant to this study (PMIDs 39753671, 38443393).

Response 1: We are grateful for this comment. Both studies have been added to the manuscript and discussed.

PMID 39753671: Bacteriophage predation is a stress factor for bacterial cells.

PMID 38443393: V. cholerae drives biofilm formation in response to a cellular component released through lysis. We sought to identify the mechanism through which V. cholerae controls biofilm formation in response to lytic phages. Reconstitution of the biofilm response to phage with mechanically produced lysate suggests that no phage component is required for the response observed in Fig. 1, and that cells commit to the biofilm state in response to a cellular factor released by lysis, which we hereafter refer to as lysis sensing. Given the pervasiveness of lysis in the environment and the diversity of lytic threats, we reason that lysis sensing could serve as a threat-agnostic mechanism that enables V. cholerae cells to detect a challenge and respond by collectively protecting themselves.

This conclusion is fully in line with our findings regarding nonspecific phage-bacteria interactions. V. cholerae study proved that bacteria are able to increase the biofilm formation in response to V. cholerae cellular components released through lysis (norspermidine), regardless of the nature of the agent causing the lysis. In contrast, P. aeruginosa tested in our experiments did not sense the mechanically produced K. pneumoniae lysate, since such sensing is probably species/genus-specific.

Question 2: At the same time both Introduction and Discussion sections may be compacted by using more concise style.

Response 2: Thank you for your suggestion regarding the conciseness of the Introduction and Discussion sections. The Introduction is currently less than one and a half pages, and the Discussion is two pages long. Other reviewers requested more detailed explanations in the revised Discussion, and they are satisfied with its final version.

Question 3:
I do not agree with the authors' rebuttal of my request to perform additional controls with purified phage stocks. The conclusion made in this paper that phage particles themselves may trigger the bacterial response is very interesting and sounds groundbreaking. However, "the outstanding claims requires outstanding proof." The argument that all the vesicles are removed by 0.22 mkm filtration doesn't sound correct for me. Those having experience with phage microscopy know well that the vesicles are annoying contaminant in many phage preps even after gradient purification (the TEM control of the purified phage is necessary!). The size of (some fraction of) the vesicles is comparable to the phage virions. To summarize, I think that this paper should not be accepted until more direct proof of the phage particles involvement in the effect is provided. Without such a proof the data present have limited novelty since they demonstrate a previously described phenomenon by a relatively known but also already published method.

Response 3: Thank you for your detailed feedback and suggestions. We understand the importance of providing robust proof for our claims. We acknowledge that vesicles can be a contaminant in phage preparations, even after gradient purification. In our study, we used nearly identical strains of Klebsiella, so their lysates are unlikely to differ significantly. 

Only one filtrate of podovirus KP34 impacted P. aeruginosa biofilm formation. The siphovirus KP36 was propagated on the host (Kp77), differing only in plasmid drug resistance genes compared to Kp486 – the host of phage KP34. Taking into account the V. cholerae study PMID 38443393 regarding cellular components released through lysis (norspermidine), we are even more convinced that our observations are correct, as the P. aeruginosa population senses KP34 virions, given that there were no changes in biofilm formation after adding a mechanically produced lysate of both Klebsiella strains.

This manuscript is a resubmission of an earlier submission. The following is a list of the peer review reports and author responses from that submission.

Round 1

Reviewer 1 Report

Comments and Suggestions for Authors

This manuscript by Drs. Gula and colleagues is quite intriguing, in that they investigate the contribution of non-host targeting phage to biofilm formation. Specifically in this study, the authors employ Klebsiella-targeting phage against Pseudomonas aeruginosa biofilms as well as one P. aeruginosa-targeting phage. Of notable interest, one of the Klebsiella-targeting phage, stimulated P. aeruginosa biofilm formation, yet had no effect on planktonic cultures. The authors conclude that general phage  detection may represent a previously undetected environmental factor for enhancing biofilm growth. I have some comments that I would like the authors to address:

1) The mechanism for assessing biofilm growth is an electrical impedance device which is described in the supplemental information. While the authors nicely describe this device, how did they assess sterility prior to experimentation?

From this reviewer's perspective, alcohol disinfection and uv exposure is not always reliable should microorganisms reside in small regions with poor chemical or uv penetration. It has been our experience to conduct a sterility assessment following disinfection, by filling a device with sterile media and allowing it to incubate for at least 24h to confirm sterility.

2) In a related point, it would be valuable to assess biofilm formation in another approach (e.g., crystal violet assay, culture based assay, or even confocal if available). The change in electrical signal may represent a change in biofilm formation (as the authors propose), or it may result in a change in  physiology, or even biofilm cell distribution. My rationale in making these suggestions is to assist the authors in their data interpretation.

3) Although speculative, do the authors have any impression of whether phage sensitivity is a general phenomenon for biofilm stimulation, or if it is limited to certain phage-bacterial systems?

Reviewer 2 Report

Comments and Suggestions for Authors

In this study, Guła and colleagues investigate the potential effects of non-specific phages on biofilm formation in Pseudomonas aeruginosa, finding that phage lysates from Klebsiella can induce biofilm formation by PAO1. This research explores a fascinating aspect of phage-bacteria interactions, especially given the ecological context where bacteria coexist with other bacterial species and their respective phages. However, to robustly claim that non-specific phages can induce biofilm formation in P. aeruginosa, additional experiments are necessary:

  • The authors appropriately use bacterial lysates as a control to determine whether the observed effects are due to internal cell components rather than the phages themselves. However, phage infection is known to activate cellular stress responses, leading to significant differences in the internal components of lysates from phage-infected versus non-infected cells. To address this, I suggest filtering the phage lysates through membranes of varying molecular weight cutoffs (e.g., 30 kDa, 10 kDa, and 3 kDa) and using the phage-free filtrates to assess their impact on biofilm production.
  • If phage-free lysates from infected cells affect biofilm production, it remains possible that phage-derived components are responsible for the observed effects. To investigate this, I recommend exposing the Klebsiella strains to other forms of stress (e.g., UV light), lysing them via sonication, and then using these lysates to assess their effect on biofilm formation. Observing an effect in this context would suggest that stress response components of Klebsiella are being sensed by P. aeruginosa.
  • Conversely, if experiments from the first point indicate that the phage itself induces biofilm production, I suggest performing adsorption assays to determine whether there is a physical interaction between the phages and the P. aeruginosa cells.

Without these additional experiments, the conclusions drawn in the manuscript, as well as the title, remain speculative. These suggested experiments will help substantiate the claim and provide a more comprehensive understanding of the interactions at play.